# INFORMATION DISTANCE FOR NEURAL NETWORK FUNCTIONS

## ABSTRACT

We provide a practical distance measure in the space of functions parameterized by neural networks. It is based on the classical information distance, and we propose to replace the uncomputable Kolmogorov complexity with information measured by codelength of prequential coding. We also provide a method for directly estimating the expectation of such codelength with limited examples. Empirically, we show that information distance is invariant with respect to different parameterization of the neural networks. We also verify that information distance can faithfully reflect similarities of neural network functions. Finally, we applied information distance to investigate the relationship between neural network models, and demonstrate the connection between information distance and multiple characteristics and behaviors of neural networks.

## 1 INTRODUCTION

Deep neural networks can be trained to represent complex functions that describe sophisticated input-output relationships, such as image classification and machine translation. Because the functions are highly non-linear and are parameterized in high-dimensional spaces, there is relatively little understanding of the functions represented by deep neural networks. One could interpret deep models by linear approximations (Ribeiro et al., 2016), or from the perspective of piece-wise linear functions, such as in (Arora et al., 2018).

If the space of functions representable by neural networks admits a distance measure, then it would be a useful tool to help analyze and gain insight about neural networks. A major difficulty is the vast number of possibilities of parameterizing a function, which makes it difficult to characterize the similarity given two networks. Measuring similarity in the parameter space is straightforward but is restricted to networks with the same structure. Measuring similarity at the output is also restricted to networks trained on the same task. Similarity of representations produced by intermediate layers of networks is proved to be more reliable and consistent (Kornblith et al., 2019), but is not invariant to linear transformations and can fail in some situations, as shown in our experiments.

In this paper, we provide a distance measure of functions based on information distance (Bennett et al., 1998), which is independent of the parameterization of the neural network. This also removes the arbitrariness of choosing "where" to measure the similarity in a neural network. Information distance has mostly been used in data mining (Cilibrasi & Vitányi, 2007; Zhang et al., 2007). Intuitively, information distance measures how much information is needed to transform one function to the other. We rely on prequential coding to estimate this quantity. Prequential coding can efficiently encode neural networks and datasets (Blier & Ollivier, 2018). If we regard prequential coding as a compression algorithm for neural networks, then the codelength can give an upper bound of the information quantity in a model.

We propose a method for calculating an approximated version of information distance with prequential coding for arbitrary networks. In this method, we use KL-divergence in prequential training and coding, which allow us to directly estimate the expected codelength without any sampling process. Then we perform experiments to demonstrate that this information distance is invariant to the parameterization of the network while also being faithful to the intrinsic similarity of models. Using information distance, we are able to sketch a rough view into the space of deep neural networks and uncover the relationship between datasets and models. We also found that information distance can

help us understand regularization techniques, measure the diversity of models, and predict a model's ability to generalize.

## 2 METHODOLOGY

Information distance measures the difference between two objects by information quantity. The information distance between two functions $f_A$ and $f_B$ can be defined as (Bennett et al., 1998):

$$d(f_A, f_B) = max\{K(f_A|f_B), K(f_B|f_A)\} \tag{1}$$

This definition makes use of Kolmogorov complexity: $K(f_B|f_A)$ is the length of the shortest program that transforms $f_A$ into $f_B$, and information distance $d$ is the larger length of either direction. (Note that this is not the only way to define information distance with Kolmogorov complexity, however we settle with this definition for its simplicity.) Intuitively, this is the minimum number of bits we need to encode $f_B$ with the help of $f_A$, or how much information is needed to know $f_B$ if $f_A$ is already known.

Given two functions $f_A : X \to Y$ and $f_B : X \to Y'$ defined on the same input space $X$, each parameterized by a neural network with weights $\theta_A$ and $\theta_B$, we want to estimate the information distance between $f_A$ and $f_B$. The estimation of Kolmogorov complexity term is done by calculating the codelength of prequential coding, so what we get is an upper bound of $d$, which we denote by $d_p$ ($p$ for prequential coding).

### 2.1 ESTIMATING $K(f_B|f_A)$ WITH PREQUENTIAL CODING

To send $f_B$ to someone who already knows $f_A$, we generate predictions $y_i$ from $f_B$ using input $x_i$ sampled from $X$. Assume that $\{x_i\}$ is known, we can use prequential coding to send labels $\{y_i\}$. If we send enough labels, the receiver can use $\{x_i, y_i\}$ to train a model to recover $f_B$.

If $f_A$ and $f_B$ have something in common, i.e. $K(f_B|f_A) < K(f_B)$, then with the help of $f_A$ we can reduce the codelength used to transmit $f_B$. A convenient way of doing so is to use $\theta_A$ as the initial model in prequential coding. The codelength of $k$ samples is:

$$L^{\text{preq}}(y_{1:k}|x_{1:k}) := -\sum_{i=1}^{k} \log p_{\hat{\theta}_i}(y_i|x_{1:i}, y_{1:i-1}) \tag{2}$$

where $\hat{\theta}_i$ is the parameter of the model trained on $\{x_{1:i-1}, y_{1:i-1}\}$, and $\hat{\theta}_1 = \theta_A$. With sufficient large $k$, the function parameterized by $\hat{\theta}_k$ would converge to $f_B$.

If both $f_A$ and $f_B$ are classification models, we can sample $y$ from the output distribution of $f_B$. In this case, the codelength (2) not only transmits $f_B$, but also $k$ specific samples we draw from $f_B$. The information contained in these specific samples is $\sum_{i=1}^{k} \log p_{\theta_B}(y_i|x_i)$. Because we only care about estimating $K(f_B|f_A)$, using the "bits-back protocol" (Hinton & van Camp, 1993) the information of samples can be subtracted from the codelength, resulting in an estimation of $K(f_B|f_A)$ as $L_k(f_B|f_A)$:

$$L_k(f_B|f_A) = -\sum_{i=1}^{k} \log p_\theta(y_i|x_{1:i}, y_{1:i-1}) + \sum_{i=1}^{k} \log p_{\theta_B}(y_i|x_i) \tag{3}$$

In practice, we want to use $k$ sufficiently large such that $f_{\hat{\theta}_k}$ can converge to $f_B$, for example by the criterion

$$\mathbb{E}_x[D_{KL}(f_B(x)||f_{\hat{\theta}_k}(x))] \le \epsilon \tag{4}$$

However, empirically we found that this often means a large $k$ is needed, which can make the estimation using (3) unfeasible when the number of available $x$ is small. Also the exact value of (3) depends on the specific samples used, introducing variance into the estimation.

## 2.2 THE PRACTICAL INFORMATION DISTANCE $d_p$

We propose to directly estimate the expectation of $L_k(f_B|f_A)$, which turns out to be much more efficient in the number of examples $x$, by leveraging infinite $y$ samples. The expectation of codelength $\mathbb{E}_{y_{1:k}}[L_k]$ over all possible samples $y_{1:k}$ from $f_B$ is:

$$\mathbb{E}_{y_{1:k} \sim f_B(x_{1:k})}[L_k(f_B|f_A)] = -\sum_{i=1}^{k} \mathbb{E}_{y_{1:i}} \log p_\theta(y_i|x_{1:i}, y_{1:i-1}) + \sum_{i=1}^{k} \mathbb{E}_{y_i} \log p_{\theta_B}(y_i|x_i)$$

$$\geq -\sum_{i=1}^{k} \mathbb{E}_{y_i} \log \mathbb{E}_{y_{1:i-1}} p_\theta(y_i|x_{1:i}, y_{1:i-1}) + \sum_{i=1}^{k} \mathbb{E}_{y_i} \log p_{\theta_B}(y_i|x_i)$$

$$= \sum_{i=1}^{k} D_{KL}(f_B(x_i)||\mathbb{E}_{y_{1:i-1}} f_{\hat{\theta}_i}(x_i)) \tag{5}$$

$$\approx \sum_{i=1}^{k} D_{KL}(f_B(x_i)||f_{\bar{\theta}_i}(x_i)) =: L'(f_B|f_A) \tag{6}$$

In (5), $\mathbb{E}_{y_{1:i-1}} f_{\hat{\theta}_i}(x_i)$ represents an infinite ensemble of models $\hat{\theta}_i$ estimated from all possible samples $y_{1:i-1}$. We replace this ensemble with a single model $\bar{\theta}_i$ that is directly trained on all the samples. $\bar{\theta}_i$ is trained using KL-divergence as objective, which is equivalent to training with infinite samples (see Appendix A for details from (5) to (6)).

The expected codelength $\mathbb{E}[L_k]$ is related, via (6), to the KL-divergence between the output distributions of $f_B$ and $f_{\bar{\theta}_i}$. Another interpretation of the above estimation is, we finetune model $\theta_A$ with an increasing number of outputs generated by $\theta_B$, and aggregate the KL-divergence between the two models along the way. The more information $f_A$ shares with $f_B$, the faster the KL-divergence decreases, resulting in a lower estimation of $K(f_B|f_A)$.

Now $d_p$ is the approximation of (1) we propose in this paper:

$$d_p(f_A, f_B) \mathrel{\hat{=}} \max\{L'(f_A|f_B), L'(f_B|f_A)\} \tag{7}$$

## 2.3 PROPERTIES OF $d_p$

The information distance $d$ in (1) applied to functions defines a metric on the space of functions. Now we check if $d_p$ satisfy the axioms of a metric:

1. $d_p(f_A, f_B) = 0 \Leftrightarrow f_A = f_B$: $f_A = f_B$ if and only if they always produce the same predictions, which is equivalent to $d_p(f_A, f_B) = 0$
2. $d_p(f_A, f_B) = d_p(f_B, f_A)$: by definition.
3. $d_p(f_A, f_B) \leq d_p(f_A, f_C) + d_p(f_C, f_B)$: whether $d_p$ keeps this property of $d$ depends on the efficiency of prequential coding, which in turn depends on model optimization.

Another important property of the information distance $d$ is the invariance with respect to the parameterization of function $f$. We found that $d_p$ is also largely invariant to parameterization of the functions. $d_p$ can be used to compare models trained differently, having different structures, or even trained on different tasks. The only condition is that both models should have sufficient expressibility to allow approximation of each other.

There is also a connection between $L_k(f_B|f_A)$ and the *information transfer* measure $L_{IT}$ (Zhang et al., 2020):

$$L_{IT}^k(\theta_n) = L_{\theta_0}^{\text{preq}}(y_{1:k}|x_{1:k}) - L_{\theta_n}^{\text{preq}}(y_{1:k}|x_{1:k})$$

as $n \to \infty$, $\theta_n \to \theta_B$, and when $y_i \sim f_B(x_i)$, we have

$$\mathbb{E}[L_{IT}^k(\theta_n)] = \mathbb{E}[L_{\theta_A}^{\text{preq}}(y_{1:k}|x_{1:k})] - \mathbb{E}[L_{\theta_B}^{\text{preq}}(y_{1:k}|x_{1:k})]$$

$$= \mathbb{E}[L_k(f_B|f_A)] \tag{8}$$

## 2.4 DATA-DEPENDENCY AND EQUIVALENT INTERPRETATIONS OF DATA

In machine learning, we often only care about the output of a model on the data distribution of the task. Neural network models are trained on input data from a specific domain, for example, image classification models take natural images in RGB format as valid input. It would be meaningless to discuss the behavior of such a model on non-RGB images. This is an important distinction between a neural network function and a function in the mathematical sense.

This motivates us to take a data-dependent formulation of distance measure. In this paper, we limit our discussion to distribution-dependent information distance:

$$d_p(f_A, f_B) = \max\{K(f'_A|f_B), K(f'_B|f_A)\} \tag{9}$$

where

$$f'_A = \underset{f \in \mathcal{F}_A}{\arg\min} \, K(f|f_B), \qquad f'_B = \underset{f \in \mathcal{F}_B}{\arg\min} \, K(f|f_A) \tag{10}$$

are equivalencies of $f_A$ and $f_B$ in the below function family ($\square$ can be $A$ or $B$):

$$\mathcal{F}_\square = \{f | \mathbb{E}_{x \sim D}[D_{KL}(f(x)||f_\square(x))] \le \epsilon\} \tag{11}$$

$\mathcal{F}_\square$ is a set containing all the functions producing outputs almost indistinguishable from $f_\square$, in the expected sense over $x$ drawn from data distribution $D$. Because they produce almost identical outputs for $x \sim D$, we call them equivalent interpretations of data in $D$. Intuitively, this means that instead of transmitting $f_B$, we can transmit $f'_B$, which is equivalent to $f_B$ on $D$, if $f'_B$ can be transmitted in fewer bits.

A quick note on why data-dependency here in the context of neural network models does not break the definition of information distance: if $f$ is a neural network trained on dataset $D$, then $f$ is fully determined by the information in $D$ plus a random seed (which is of negligible information).

By introducing data-dependency, it enables us to approximate Kolmogorov complexity by coding samples drawn from data distribution $D$, in other words we can use the training set for coding.

## 3 EMPIRICAL STUDY

The proposed information distance $d_p$ relies on an estimation of Kolmogorov complexity of neural networks with prequential codelength, which unfortunately does not have known theoretical guarantees. Therefore we validate the performance of $d_p$ mainly with empirical results. We use experiments to show the advantages of $d_p$ and in what situations are $d_p$ useful.

### 3.1 EXPERIMENT SETUP

All experiments in this section are performed using ResNet-56 models (He et al., 2016) on Tiny-ImageNet[1], a 200-class image classification dataset. To make the codelength estimation more reliable, we try to achieve lower codelength in prequential coding by optimizing the model estimation process in sequential coding. We performed a hyper-parameter search to select optimal hyper-parameters that result in the lowest codelength. Unless otherwise stated, we use $k = 10000$ in experiments throughout this paper, which we found in most cases allow the difference between coding model and the reference model $\mathbb{E}_x[D_{KL}(f_B(x)||f_{\hat{\theta}_k}(x))]$ to converge.

### 3.2 INVARIANCES OF INFORMATION DISTANCE

A prominent advantage of information distance is independent of parameterization. Neural networks like multi-layer perceptrons can have a very large number of different configurations (units, weights) that correspond to the same function. Because there is no "canonical" way of parameterizing neural networks, comparing the input-output functions represented by different neural networks can be difficult by merely looking at the network parameters.

There exist many metrics for measuring the similarity or distance between neural networks. But because neural networks are so versatile, similar neural networks can look similar under some metric

---

[1]https://tiny-imagenet.herokuapp.com

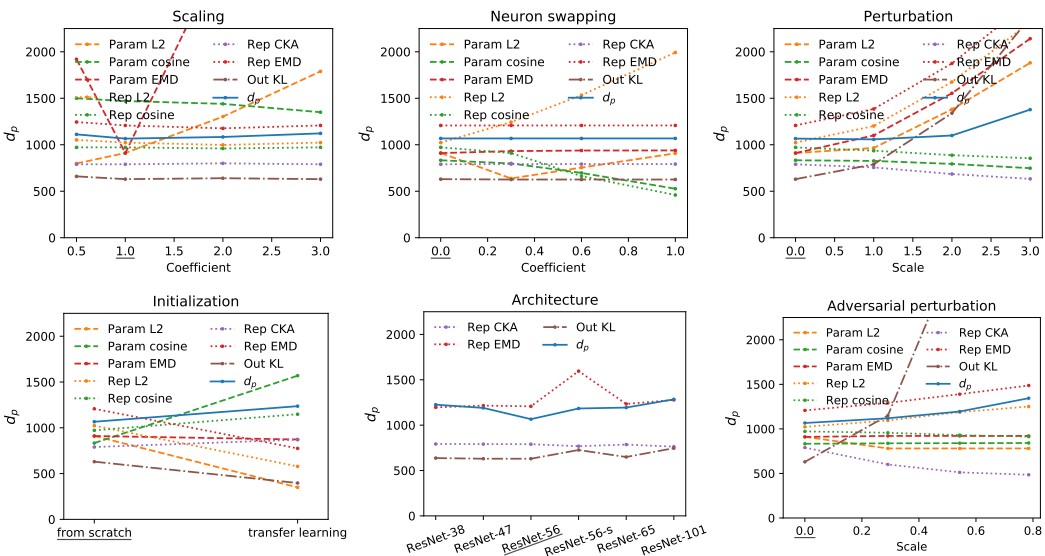

Figure 1: Distance $d_p$ changes with respect to changes in the parameterization of the networks. We measure the distance between a pair of ResNet-56 models $\theta_A$ and $\theta_B$, trained on two subsets of Tiny-ImageNet, respectively. We modify the configuration of $\theta_A$ while keeping $\theta_B$ the same. Modifications include: **scaling**, where we multiply the weights of the network by a coefficient $c$, **neuron swapping**, where we randomly permute a fraction $c$ of the units within a layer, **perturbation**, where Gaussian random noise of zero mean and standard deviation $c$ is added to the weights of the network, and **adversarial perturbation**, where a vector of standard deviation $c$ is added to the weights of the network to maximize the change of second-to-last layer representations. We also include training $\theta_A$ with different **initializations** and using different network **architectures**. Some commonly used distance measures are used as baselines. These include measures in three different spaces: **parameter space distances**, which measure distance using metrics on the parameter matrices, including plain L2 distance, cosine similarity, and Earth Mover Distance (EMD) (Monge, 1781; Rubner et al., 1998) that computes the cost to align neurons. **Representation space distances** measure distance on the second-to-last layer representation vectors, including plain L2, cosine, the EMD cost of aligning feature dimensions, and Linear Centered Kernel Alignment (CKA) (Kornblith et al., 2019) which is based on pairwise sample similarity and outperforms previously proposed similarity measures. For **output space distance**, we use the KL-divergence between the probability distribution generated by the final softmax layer. The baseline measures include both common and straightforward measures and more sophisticated measures like EMD and CKA that possess invariance properties. Underlined labels on the $x$-axis denote the distances measured between unmodified $\theta_A$ and $\theta_B$. Curves for each measure are individually scaled (only scaled, not shifted) to ease viewing on the same graph. Cosine similarity and CKA value are within [0,1] and is inversely correlated with distance.

while very dissimilar by other metrics. There lacks a universal definition of similarity, which is precisely the problem we try to solve with $d_p$.

To empirically examine the invariance of $d_p$, we evaluated $d_p$ under different re-parameterizations of a neural network. We also include a number of distance metrics as baselines for comparison. Descriptions of test scenarios, baselines, and the results are shown in Figure 1. Table 1 summarizes the observed invariance of distance measures with a quantitative measure.

Results indicates that $d_p$ is relatively stable under different kinds of re-parameterization of the network and is the most invariant overall. Other distance measures all exhibit strong dependency on certain kind of parameterization or is inapplicable for some parameterizations. For re-parameterizations that does not change or only minimally change the function $f$ (scaling, neuron swapping, initialization, architecture), $d_p$ also exhibit minimal change. For adding perturbations to the network, information distance only starts to increases when the perturbation is large enough. This is because only large noise will start to "wipe out" information in the network. $d_p$ is also robust to small adversarial perturbations, while also showing that adversarial perturbations destroy information in the network faster than random noise.

Table 1: Measuring invariancy of distance measures. Invariancy is measured by the relative change of distance in each test scenario (averaged over all data points). Lower value means more invariant. Oracle refers to an ideal distance measure. N/A means the method cannot be used in that test.

| Space | Method | Scaling | Neuron swapping | Perturbation | Adversarial perturbation | Initialization | Architecture | Mean |
|---|---|---|---|---|---|---|---|---|
| Parameter | L2 | 0.51 | 0.16 | 0.55 | 0.14 | 0.62 | n/a | 0.39 |
| | Cosine | 0.04 | 0.19 | 0.16 | 0.02 | 0.88 | n/a | 0.26 |
| | EMD | 2.39 | 0.03 | 0.76 | 0.01 | 0.04 | n/a | 0.65 |
| Representaiton | L2 | 0.02 | 0.55 | 0.69 | 0.15 | 0.43 | n/a | 0.37 |
| | Cosine | 0.00 | 0.30 | 0.08 | 0.04 | 0.18 | n/a | 0.12 |
| | EMD | 0.02 | 0.00 | 0.60 | 0.15 | 0.36 | 0.08 | 0.20 |
| | Linear CKA | 0.00 | 0.00 | 0.12 | 0.33 | 0.10 | 0.01 | 0.10 |
| Output | $D_{KL}$ | 0.02 | 0.01 | 1.38 | 3.59 | 0.37 | 0.08 | 0.91 |
| Other | $d_p$ | 0.04 | 0.00 | 0.11 | 0.14 | 0.07 | 0.03 | **0.06** |
| | *Oracle* | $\sim 0$ | 0 | $>0, \ll 1$ | $>0, \ll 1$ | $\sim 0$ | $\sim 0$ | - |

In *Initialization* experiments, we initialize $\theta_A$ with a network pre-trained on CIFAR-10 (Krizhevsky & Hinton, 2009). $d_p$ increase 7% compared to random initialization, because $\theta_A$ carries over some information from CIFAR-10, making $\theta_A$ and $\theta_B$ slightly less similar. In *Architecture* experiments, if $\theta_A$ uses a different architecture than $\theta_B$ (which is ResNet-56), we also observe increase in $d_p$. The more different the models are (in terms of number of layers) form ResNet-56, the distance $d_p$ is also slightly higher. This indicates that while $d_p$ is largely invariant to model parameterizations, it is also consistent with intuitive similarities of models. This is not observed with EMD and CKA distances.

### 3.3 IS INFORMATION DISTANCE FAITHFUL?

Having seen the invariance properties of $d_p$, we next check if $d_p$ is faithful to reflect true model distance. Empirically, we verify if $d_p$ is aligned with intuitive indicators of model distance. We experiment with two settings: examining model interpolation and the model training progress. These can serve as a sanity-check that $d_p$ does reflect differences in the information of models.

In *model interpolation* we use a straightforward way to manipulate the distance between models: we use two ResNet-56 models $\theta_A$ and $\theta_B$ trained on Tiny-Imagenet, and perform linear interpolation in parameter space to get model $\theta = (1 - c)\,\theta_A + c\,\theta_B$. As $c$ change from 0 to 1, the model $\theta$ goes smoothly from $\theta_A$ to $\theta_B$. We apply $d_p$ to measure the distance from the interpolated model $\theta$ to $\theta_B$ for different $c$. The results are shown in Figure 2.[2]

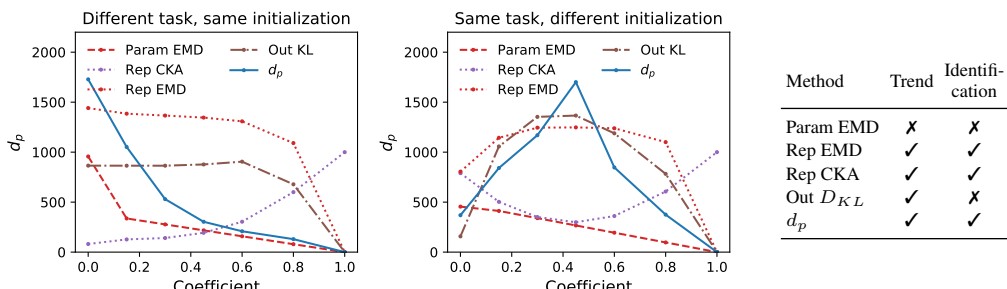

Figure 2: Distance from the interpolation model $\theta$ to $\theta_B$. Left: $\theta_A$ and $\theta_B$ are trained on different but related tasks (first 100-class and last 100-class of Tiny-ImageNet) from the same initialization. Middle: $\theta_A$ and $\theta_B$ are trained on the same task but from different initializations. Right: whether the measure's trend is correct, and whether it can be used to identify different interpolation coefficients.

As we interpolate two functions $f_A$ and $f_B$ in the parameter space, if $\theta_A$ and $\theta_B$ are parameterized similarly, we observe $d_p(f, f_B)$ to monotonically decrease as $f$ gets closer to $f_B$ (Figure 2 left). At the beginning when $c$ is small, increase in $c$ introduces more "fresh information" about $f_B$, thus $d_p$

---

[2]To avoid clutter in graphs, we did not include L2 and cosine measures in Figure 2 and 3, as they fail basic invariancy tests in Section 3.2. Full results are given in Appendix B.2.

decreases faster than later in interpolation. On the other hand, if we interpolate two functions that are parameterized differently, because linear mixing of $\theta_A$ and $\theta_B$ in parameter space leads to a degraded network, the distance would first increase then decrease, indicating a loss of information middle in the interpolation. Overall, the general trend of $d_p$ agrees with advanced similarity measures like representation EMD and CKA in this scenario.

In the experiment *training progress*, we use $d_p$ to examine the training progress of a model. Starting from $\theta_0$, we train the model $n$ epochs to converge and measure the distance from the $i$-th epoch model $\theta_i$ to the initial model $\theta_0$ as well as to the final model $\theta_n$. Results are shown in Figure 3.

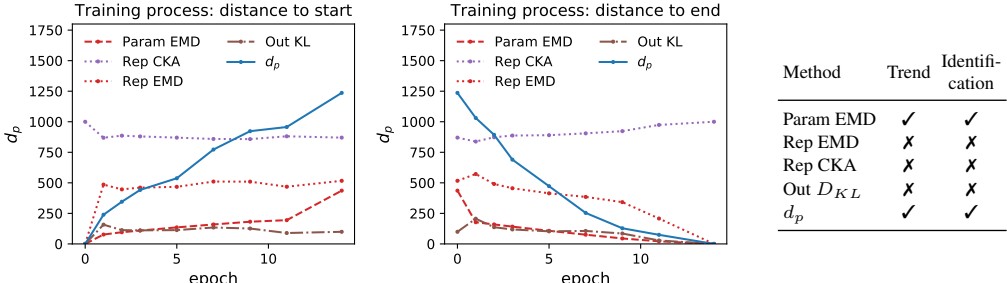

Figure 3: Distance from the $i$-th epoch model $\theta_i$ to the initial model $\theta_0$ (left) and the final model $\theta_{14}$ (middle). Right: whether the measure has monotonic trend with respect to the training progress, and whether it can be used to identify models from different epochs.

With the training progress after each epoch, $d_p(f_i, f_0)$ steadily decreases and $d_p(f_i, f_n)$ increases. Change in distance is faster in earlier epochs because most of the learning happens in the first few epochs. In this scenario, $d_p$ is monotonic with training and correctly depicts the training progress. Both representation EMD and CKA fail to show the dynamics of training beyond the first epoch.

To summarize, parameter space distances fail when function similarity does not correspond to parameter value similarities, and representation space distances can be too noisy to be reliable when similarity is high. Only information distance $d_p$ remains faithful in both scenarios.

## 4 APPLICATION

To illustrate the utility of a universal function distance, we provide a few scenarios where we use $d_p$ for understanding and making predictions.

### 4.1 SKETCHING THE GEOMETRY OF DATA AND MODEL SPACE

A distance measure can help us understand the relationship between datasets and between models. Datasets and models usually live in very high-dimensional spaces, which makes it hard to directly perform a comparison. Instead, we can use $d_p$ to get the information distance between datasets and models. In computer vision there is a myriad of datasets and model structures, and we use the Visual Task Adaptation Benchmark (VTAB) (Zhai et al., 2019) as a collection of vision datasets. On each dataset, a model is trained to represent the input-output function of the task. Then we use $d_p$ to measure pairwise distances between these functions. To help to visualize the relationship, we use Isometric Mapping (Tenenbaum et al., 2000), a manifold learning algorithm to generate three-dimensional embeddings for each function. The distance of points in three-dimensional space is optimized to keep the original structure.

Distances can tell a lot about the relationship between models. In the nine large datasets of VTAB, datasets cluster largely according to the three categories proposed in VTAB (natural, specialized, and structured). CIFAR-100 is very different from any other datasets, but is relatively closer to satellite image datasets than to artificial shape datasets. SVHN (Netzer et al., 2011) is close to 2D shape datasets. The four small datasets are evenly distributed in space: no pair of them is very similar. In terms of model architecture, ResNet variants are relatively similar, while AlexNet (Krizhevsky, 2014) and VGG (Simonyan & Zisserman, 2014) are farther away. VGG with batch normalization

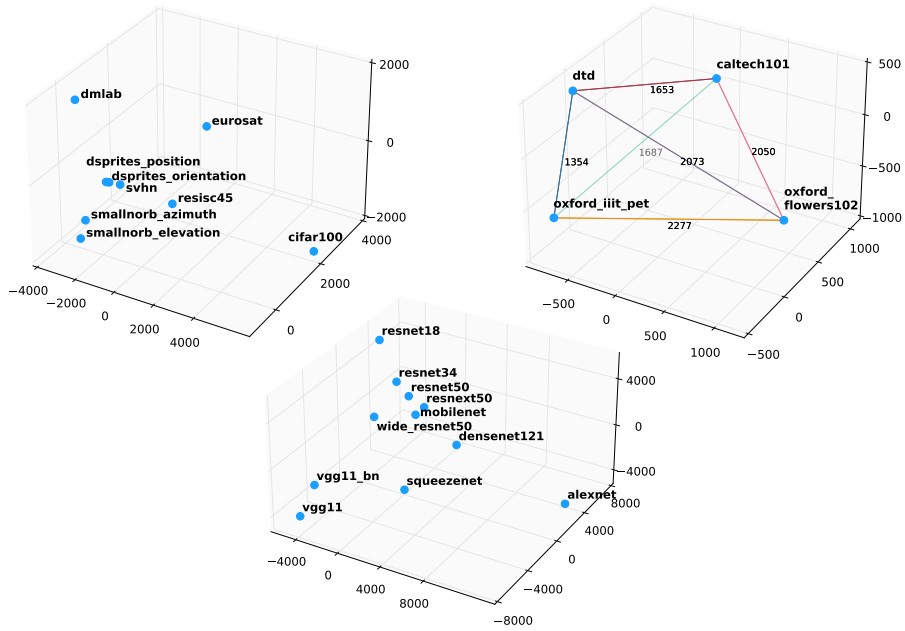

Figure 4: Visualizing distances between datasets and models in three-dimensional space. Top-left: large datasets in VTAB. Top-right: small datasets in VTAB. Bottom: various model architectures trained on ImageNet. The numbers on colored lines are pairwise distances.

is closer to ResNet than without. ResNet-50, ResNeXt-50 (Xie et al., 2017) and WideResNet-50 (Zagoruyko & Komodakis, 2016) are closest as they are indeed very similar.

## 4.2 UNDERSTANDING REGULARIZATIONS

Regularization techniques like L2 regularization can bias the learned neural networks toward less complex functions, while for techniques like dropout (Srivastava et al., 2014) and self-distillation (Furlanello et al., 2018), the regularization effect may be less straightforward to explain. We can use $d_p$ to examine the (information) complexity of a network $f$, by measuring its distance $d_p(f, \mathbf{0})$ to an empty function.

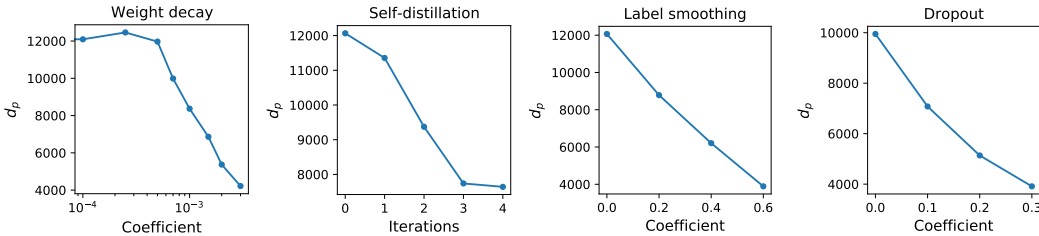

Figure 5: Distance to an empty function $d_p(f, \mathbf{0})$ for models with different kinds of regularization and varying strength. From left to right: Weight decay (L2 regularization), self-distillation, label smoothing, dropout. For dropout, a different base model without batch normalization is used.

From Figure 5, we observe that all the listed techniques result in a reduction of $d_p(f, \mathbf{0})$, which means that the information complexity of the model function $f$ is reduced. For weight decay, information complexity only starts to decrease after the regularization coefficient is larger than a threshold. Self-distillation has a similar effect to regularization, with the number of distillation iterations controlling regularization strength. This agrees with the theoretical analysis in (Mobahi et al., 2020). Label smoothing and dropout also result in simpler models, highlighting their regularization effect.

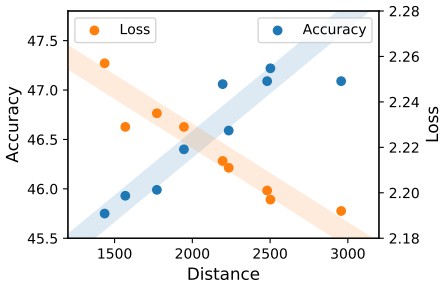 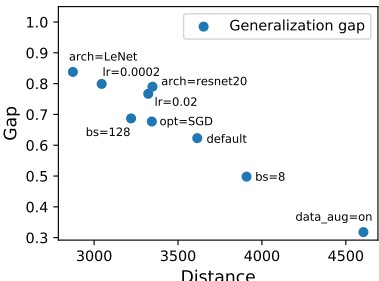

Figure 6: Relation between ensemble performance and model diversity given by $d_p(f_1, f_2)$.

Figure 7: Relation between generalization gap and model complexity $d_p(f, \mathbf{0})$.

### 4.3 ENSEMBLES AND MODEL DIVERSITY

Distance can be used as an indicator of model diversity: the larger $d_p$ between models, the more diverse the models are. Ensembling is a common technique to use the consensus of multiple models to deliver superior performance than a single model. We speculate that a larger model diversity will result in more performance gain from ensembles.

To verify this connection, we train a number of models on Tiny-Imagenet, all to the same performance on the validation set, but with different initializations and different subsets of the training set. Then we choose models in pairs to measure their ensemble performance as well as the distance $d_p(f_1, f_2)$ between them. The result is given in Figure 6, and we found a clear correlation between $d_p$ and ensemble performance, and the relationship is about linear. This also indicates that $d_p$ captures model diversity.

### 4.4 PREDICTING GENERALIZATION

Finally, $d_p$ is also linked with model generalization. Generalization of neural networks is heavily affected by hyper-parameters and optimization. There have been several works aiming to find the relationship between generalization performance and properties of the network, but it turns out that predicting generalization gap can be a challenging task (Jiang et al., 2019; 2020).

We perform a small-scale experiment to illustrate the connection between information distance and generalization gap. We train a number of models with different hyper-parameters (batch size, learning rate, optimizer, etc.) all to the same loss on the training set, and then measure the distance to a random model by $d_p(f, \mathbf{0})$. In Figure 7, we observe that the information complexity of the model is also linked with generalization gap, which also turns out to be a roughly linear relationship. Models that generalize better are farther away from a random model than less performing models.

## 5 DISCUSSION AND CONCLUSION

The proposed distance $d_p$ is based on information distance defined with Komolgorov complexity $K$. We do not attempt to give a good estimation of $K$, but instead relying on the efficiency of prequential coding, we empirically illustrate that $d_p$ share the invariance properties of information distance, and reflects the similarity relationships of functions parameterized by neural networks. We also found that $d_p$ is linked with behaviors of models, making it a potential tool for analyzing neural networks.

The most notable difference between $d_p$ and other similarity metrics is universality. Theoretically rooted in information distance, $d_p$ is independent of parameterization and widely applicable in situations involving different tasks and models. However, $d_p$'s utilization of prequential coding also introduces limitations that it might not work in situations where prequential coding fails, for example, when $f$ cannot be efficiently approximated by neural networks.

$d_p$ could introduce a potential scale-free, or even parameterization-free geometry of space spanned by neural models. Optimization with manifold descent by $d_p$ could also remove the dependency on parameterization, thus avoiding ill-posed conditions in some parameterizations (Dinh et al., 2017).

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

## A  TECHNICAL DETAILS IN CALCULATING $d_p$

In equations (5)-(6), we used $f_{\bar{\theta}_i}$ in place of the ensemble model $\mathbb{E}_{y_{1:i-1}} f_{\hat{\theta}_i}(x_i)$

$$\mathbb{E}_{y_{1:k} \sim f_B(x_{1:k})}[L_k(f_B|f_A)] \geq \sum_{i=1}^{k} D_{KL}(f_B(x_i)||\mathbb{E}_{y_{1:i-1}} f_{\hat{\theta}_i}(x_i)) \tag{12}$$

$$= \sum_{i=1}^{k} D_{KL}(f_B(x_i)||f_{\bar{\theta}_i}(x_i)) + \mathbb{E}_{y_i \sim f_B(x_i)} \log \left( \frac{\mathbb{E}_{y_{1:i-1}} f_{\hat{\theta}_i}(x_i)}{f_{\bar{\theta}_i}(x_i)} \right) \tag{13}$$

$\bar{\theta}_i$ is trained with objective function $c_{KL}$:

$$c_{KL}(\theta) = \sum_{j=1}^{i} D_{KL}(f_B(x_j)||f_\theta(x_j)) \tag{14}$$

$$= \sum_{j=1}^{i} [H_c(f_B(x_j), f_\theta(x_j)) - H(f_B(x_j))] \tag{15}$$

$$= \mathbb{E}_{y_{1:i} \sim f_B(x_{1:i})} \sum_{j=1}^{i} H_c(y_j, f_\theta(x_j)) - \sum_{j=1}^{i} H(f_B(x_j)) \tag{16}$$

$$= \mathbb{E}_{y_{1:i} \sim f_B(x_{1:i})} c_{CE}(\theta) - \sum_{j=1}^{i} H(f_B(x_j)) \tag{17}$$

where $c_{CE}$ is the cross-entropy objective function used to train $\hat{\theta}_i$. $H_c$ stands for cross-entropy.

In other words, $\bar{\theta}_i$ is trained with the average loss used in training $\hat{\theta}_i$ (the entropy term in (17) does not depend on $\theta$ and has no effect in training). Therefore $f_{\bar{\theta}_i}$ should mimic the behavior of the infinite ensemble $\mathbb{E}_{y_{1:i-1}} f_{\hat{\theta}_i}(x_i)$ reasonably well and make the second term in (13) small.

Using (6) instead of (3) to estimate codelength not only makes estimations independent of the sampling process, but also requires fewer input examples $x$. This is because we are making the most use of each $x$ by essentially drawing infinite $y$ samples from each $f(x)$. Generally speaking, to estimate $d_p$, one first needs to sample some input $x$ from $D$ (based on data-dependency introduced in Section 2.4). Usually $D$ is unknown, but we have a dataset $S$ containing samples from $D$ so that we can use examples from $S$ instead. When the size of $S$ is small, there may not be enough examples to train $f_A$ to converge to $f_B$ by (3). We found that this is often the case for small datasets, for example in Section 4.1. Even when $S$ is large, we can save computation time by using a smaller sample size $k$.

In Section 2.4 we introduced data dependency, where we study function restricted to the data distribution $D$. We can quickly see that if $f$ represents a model trained on data distribution $D$, then

$$K(f|f', D) = K(f'|f, D) = 0$$

where $f'$ means $f$ restricted to $D$. Because $f$ is fully determined by $f'$. And

$$K(f'_B|f'_A, D) = K(f_B|f_A, D)$$

follows from above. We can study $K(f'_B|f'_A, D)$ by coding examples from $D$. This requires the functions to be compared ($f_B$ and $f_A$) trained on the same kind of input. This is a reasonable restriction because it is unlikely one would be concerned about the distance between functions defined on different input spaces.

## B  EXPERIMENT DETAILS

### B.1  INVARIANCE EXPERIMENTS

We provide details for re-parametreizations used in invariance experiments:

- Scaling: the weights in layer $i$ is multiplied by $c$, and the weights in layer $i-1$ multiplied by $1/c$. For relu networks, this keeps the output of network unchanged.
- Neuron swapping: we randomly permute $c \cdot$(total number of neurons) neurons in layer $i$. We also correspondingly permute the input of layer $i+1$ so that the network output is unchanged.
- Perturbation: we add Gaussian noise of zero mean and standard deviation $c$ to each individual weight of the network.
- Adversarial perturbation: we add a vector $v$ of standard deviation $c$ to the weight vector of the network, and optimize the vector to maximize deviations of second-to-last layer representations. i.e. $max_{v:std(v)=c}\mathbb{E}_x[(f_\theta^r(x) - f_{\theta+v}^r(x))^2]$
- Initialization: we experiment with random initialization and initialize with a pre-trained network on another dataset (CIFAR-10).
- Architecture: we use ResNet architecture with different width and depth. ResNet-56-s refers to ResNet-56 with half the width in each layer.

Next we list the baseline distance (or similarity) measures and describe how to calculate them for network $f_A$ and $f_B$:

Parameter space distances: we denote the $i$-th layer weight matrix of network $f_A$ as $w_A^i$. For L2 and cosine measures, we first flatten and concatenate all weight matrices $w_A^i$ and biases of the network into a long vector $w_A^{all}$ (excluding parameters in batch normalization layers, because some statistic variable in them can be large and dominate the norm of vector $w$).

- L2: $d_{l2} = ||w_A^{all} - w_B^{all}||_2$.
- Cosine: $d_{cosine} = w_A^{all} \cdot w_B^{all}/(||w_A^{all}||_2||w_B^{all}||_2)$.
- EMD: we use "Optimal Transport of Neurons" in (Li et al., 2020). The distance matrix $M$ is taken to be the pairwise L2 distance between weights of each neuron

$$M_{mn}^i = ||w_{Am\cdot}^i - w_{Bn\cdot}^i||_2$$

  The EMD distance is the optimal transport cost of matching neurons from one network to neurons in the other network

$$d_{emd} = \min_{P \in \Pi(\mu,\nu)} \langle P, M \rangle_F$$

  where $P$ is the optimal transport plan.

Representation space distances: we sample $x_i$ from data distribution $D$ and denote the output representation vector of the second-to-last layer of model $f_A$ by $f_{Ai}$.

- L2: $d_{l2} = \frac{1}{k}\sum_{i=1}^k ||f_{Ai} - f_{Bi}||_2$.
- Cosine: $d_{cosine} = \frac{1}{k}\sum_{i=1}^k[f_{Ai} \cdot f_{Bi}/(||f_{Ai}||_2||f_{Bi}||_2)]$.
- EMD: same as in "parameter space distances", except that the distance matrix $M$ is taken to be the pairwise L2 distance between the activation vector of each neuron

$$M_{mn} = (\sum_{i=1}^k (f_{Aim} - f_{Bin})^2)^{1/2}$$

- Linear CKA: we use implementation provided by (Kornblith et al., 2019):

$$d_{cka} = \frac{||f_{B\cdot}{}^T f_{A\cdot}||_F^2}{||f_{A\cdot}{}^T f_{A\cdot}||_F ||f_{B\cdot}{}^T f_{B\cdot}||_F}$$

  where $f_{A\cdot}$ is a a matrix whose $k$ rows are vectors $f_{A1}, ..., f_{Ak}$.

Output space distances: we use the output distributions of $f_A$ and $f_B$.

- KL-divergence: $\mathbb{E}_x[D_{KL}(f_A(x)||f_B(x))]$

## B.2 MORE RESULTS OF SECTION 3.3

Figure 8 and 9 shows the results in model interpolation experiments and training progress experiments, for all distance measures studied in this paper. We also show that if each method gives the correct trend, and whether it can be used to identify different models.

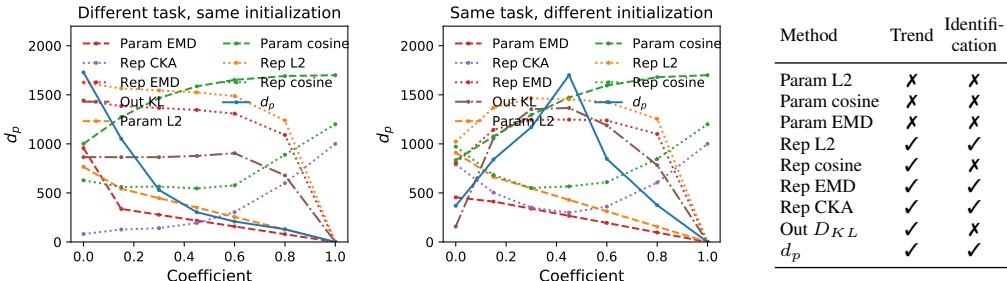

Figure 8: Full results of Figure 2: Distance from the interpolation model $\theta$ to $\theta_B$.

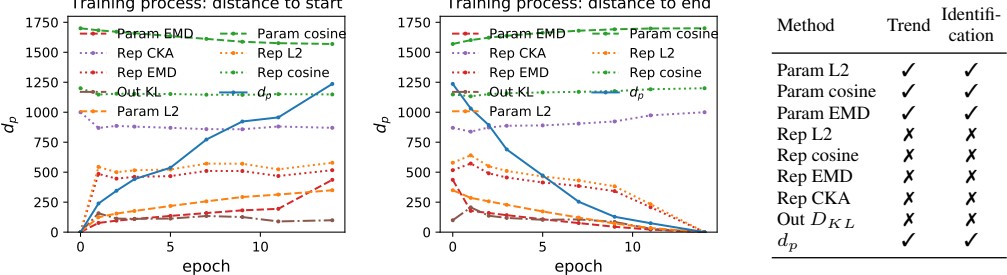

Figure 9: Full results of Figure 3: Distance from the $i$-th epoch model $\theta_i$ to initial model $\theta_0$ and final model $\theta_{14}$.

## B.3 GEOMETRY EXPERIMENTS

From the 19 datasets included in VTAB (Zhai et al., 2019), we were able to download 13 datasets for using in this work. Because the dataset size vary greatly among the 13 datasets, we divide them into two groups: larger datasets (size $> 10000$), which include:

- *cifar100*: CIFAR-100 (Krizhevsky & Hinton, 2009)
- *svhn*: SVHN (Netzer et al., 2011)
- *eurosat*: EuroSAT (Helber et al., 2019)
- *resisc45*: Resisc45 (Cheng et al., 2017)
- *dsprites_position*: dSprites/location (Matthey et al., 2017)
- *dsprites_orientation*: dSprites/orientation
- *smallnorb_azimuth*: SmallNORB/azimuth (LeCun et al., 2004)
- *smallnorb_elevation*: SmallNORB/elevation
- *dmlab*: DMLab (Beattie et al., 2016)

and smaller datasets (size $< 10000$), which include:

- *caltech101*: Caltech101 (Li et al., 2006)

- *dtd*: DTD (Cimpoi et al., 2014)

- *oxford_flowers102*: Flowers102 (Nilsback & Zisserman, 2008)

- *oxford_iiit_pet*: Pets (Parkhi et al., 2012)

For larger datasets, we use $k = 10000$ as with other experiments. For smaller datasets, we use $k = 2000$. Model is ResNet-56 trained from scratch.

In model geometry experiments, we use the following models trained on ImageNet, as provided by torchvision[3]:

- *resnet18*: ResNet-18 (He et al., 2016)

- *resnet34*: ResNet-34

- *resnet50*: ResNet-50

- *vgg11*: VGG-11 without batch normalization (Simonyan & Zisserman, 2014)

- *vgg11_bn*: VGG-11 with batch normalization

- *alexnet*: AlexNet (Krizhevsky, 2014)

- *resnext50*: ResNeXt-50-32x4d (Xie et al., 2017)

- *wide_resnet50*: WideResNet-50-2 (Zagoruyko & Komodakis, 2016)

- *densenet121*: Densenet-121 (Huang et al., 2017)

- *squeezenet*: SqueezeNet 1.1 (Iandola et al., 2016)

- *mobilenet*: MobileNet V2 (Sandler et al., 2018)

Codelength is calculated on the training set of ILSVRC 2012 (Russakovsky et al., 2015).

## B.4 Ensemble experiments

We train multiple ResNet-56 models with 2 different random initializations and half of the examples sampled form the TinyImageNet training set. This means the training examples seen by each model can overlap from 0% to 100%. We then select two models out of this collection and ensemble the two models. Ensemble performance and distance between the two models measured by $d_p(f_A, f_B)$ is given in Table 2. We list pairs with same or different initialization, and with different overlap in training examples. Generally speaking, the less the training examples overlaps the larger the distance between the models. Different initialization can also makes the models more dissimilar. Note that from Figure 6, we see that distance $d_p(f_A, f_B)$ is correlated with ensemble performance regardless of whether diversity comes from difference in training examples or difference in initializations.

Table 2: Details of configurations in ensemble experiments

| Initialization seed | Training set overlap | Ensemble loss | Ensemble accuracy | $d_p(f_A, f_B)$ |
|---|---|---|---|---|
| same | 0% | 2.197 | 47.22 | 2502 |
| same | 25% | 2.211 | 46.59 | 2234 |
| same | 50% | 2.229 | 46.40 | 1946 |
| same | 75% | 2.229 | 45.93 | 1569 |
| different | 0% | 2.192 | 47.09 | 2957 |
| different | 25% | 2.201 | 47.09 | 2481 |
| different | 50% | 2.214 | 47.06 | 2195 |
| different | 75% | 2.235 | 45.99 | 1772 |
| different | 100% | 2.257 | 45.75 | 1436 |

[3]https://github.com/pytorch/vision

## B.5 GENERALIZATION EXPERIMENTS

We run experiments on CIFAR-10 with different hyperparameters and model configurations, and in all configurations we train the model to cross-entropy loss of 0.1 on training set. Then we measure the generalization gap as the loss on testing set minus the loss on training set. Starting from a default configuration (which is the same hyperparameters we use in other experiments in this paper), each time we modify one of the hyperparameters. Results are listed in Table 3. In terms of studying generalization gap, our experiments is far less thorough than in (Jiang et al., 2020), but here we would like to illustrate the connection between $d_p$ with generalization gap under different experiment settings, without spending too many machine hours.

Table 3: Details of configurations in generalization experiments.

| Configuration | Architecture | Optimizer | Batch size | Learning rate | Data augmentation | Generalization gap | $d_p(f, \mathbf{0})$ |
|---|---|---|---|---|---|---|---|
| default | ResNet-56 | Adam | 32 | 0.002 | off | 0.32 | 4605 |
| lr=0.0002 | ResNet-56 | Adam | 32 | **0.0002** | off | 0.80 | 3045 |
| lr=0.02 | ResNet-56 | Adam | 32 | **0.02** | off | 0.77 | 3322 |
| bs=8 | ResNet-56 | Adam | **8** | 0.002 | off | 0.50 | 3908 |
| bs=128 | ResNet-56 | Adam | **128** | 0.002 | off | 0.69 | 3220 |
| opt=SGD | ResNet-56 | **SGD** | 32 | 0.2 | off | 0.68 | 3344 |
| arch=resnet20 | **ResNet-20** | Adam | 32 | 0.002 | off | 0.79 | 3348 |
| arch=LeNet | **LeNet-5** | Adam | 32 | 0.002 | off | 0.84 | 2873 |
| data_aug=on | ResNet-56 | Adam | 32 | 0.002 | **on** | 0.62 | 3615 |

