# OpenReview forum: "Information distance for neural network functions"
_ICLR.cc/2021/Conference — Reject_

### Official Review · AnonReviewer1 · 2020-10-27
**Defining information distance by relaxing Kolmogorov complexity**

**Rating:** 5
**Confidence:** 4

**Review:**

Summary:

The authors provide a practical distance measure among different neural networks. They extend the classical information distance by replacing the uncomputable Kolmogorov complexity in terms of code length of prequential coding. Empirically, they show several practical advantages of the proposed distances.

Pros: The proposed distance may be easy to estimate.

Cons:

1. For the proposed information distance definition and invariance property, are there any concrete examples or analytical formulas to demonstrate its effectiveness? E.g., if a simple linear function gives the neural network, what is the concrete value of this distance.

2. The distance is data-dependent. Suppose the data is given by a particularly known distribution, even Gaussian,
is there a mathematical way to demonstrate the defined quantities?

Some sentences need to be revised:

`"` Information distance dp is based on information distance defined with Komolgorov complexity K."

Some questions:

1. In literature, there are already many works in studying the distance and geometry associated with the neural networks, such as Fisher information geometry (S. Amari) and Wasserstein information geometry (W. Li). What is the relation between this new distance based on prequential coding's codelength and two distances mentioned above? Especially the role that KL-divergence plays in this framework?

Li, Zhao, Wasserstein information matrix.

Amari, Matsuda, Wasserstein statistics in one-dimensional location-scale model.


2. Fisher information metric is known to be invariant under parameterization and is even characterized as the unique metric on probability space to have this property. In your paper, you mention that this new practical distance is also invariant under parameterization. Does this have something to do with the classical Fisher metric?

3. To obtain the data-dependency definition of information distance, do you have to calculate an empirical version of it? If this is the case, what is the convergence properties, such as the convergence rate of this empirical information distance?

---

> ### Author Response · Authors · 2020-11-20
> **Response to Reviewer1, clarification of relationship to Fisher information and Wasserstein information**
>
> Thanks for your comments, which are very valuable suggestions.
>
> Concrete examples or analytical formulas for simple cases: it would be beneficial to have such examples, but we are not aware of existing theoretical results on the prequential codelength applied to neural networks. One potential reason is the uncomputability of Kolmogorov complexity. Our goal in this paper is mainly about providing a methodology and present empirical results. This suggestion could be an important future work.
>
> Wording of the sentence: we have revised in this version.
>
> Relationship with Fisher information and Wasserstein information: to put it simply, the problem we are studying is different. Fisher information and Wasserstein information are useful tools to study the distance between distributions, and the associated geometry of parameterized models. They study distance in the space of probability distributions over X.
>
> On the other hand, information distance studies distance in the hypothesis space H, and distance is measured with Kolmogorov complexity K. Consider two identically-shaped Gaussian distributions centered at a and -a, they can have high distances by KL-divergence or Wasserstein distance in the space of distributions, but can have a very small distance of 1 bit measured by K in the hypothesis space, because a Turing machine can use 1 bit of information to determine the sign of the mean parameter. Prequential codelength as an approximation of Kolmogorov complexity can measure “how much information is needed to transform hypothesis $f_A$ into $f_B$,” which is what we are concerned about.
>
> This is the reason we propose to use information distance in the hypothesis space instead of any distance in the space of output distributions, because very close-related models can generate very different $p(y|x)$ distributions. For example, two identical pre-trained ImageNet classifiers have their final layer finetuned on MNIST and CIFAR-10 respectively, their output distributions are not even defined on the same sample space, therefore there is no sensible way to compare the two distributions. Yet the two models are almost the same except weights in the final layers, and their information only differs by a small amount.
>
> Data-dependency and convergence properties: we are essentially calculating an empirical version of information distance throughout this paper and trying to verify its effectiveness with experiments. For the same reason as analytical formulas, unfortunately we are unable to provide convergence guarantees at this point. Currently we aim to use empirical results in the paper to justify the use of information distance in the absence of theoretical guarantees, and in the future we will put more investigation into theoretical properties of (the empirical version of) information distance.

---

### Official Review · AnonReviewer3 · 2020-10-28
**Interesting in topics, Weak in techicals, Confusing in contribution**

**Rating:** 4
**Confidence:** 4

**Review:**


The authors proposed a distance measure to characterize the information space among neural networks. They derive this measure by empirically compute the expected code length. They applied the distance measure to neural network models in didfferent settings and compared with some baselines.


Strength:

+ the information theoretic perspective of the deep learning is a useful approach to compare neural networks.
+ the spectrum of the empirical study is pretty wide, covering some interesting aspects of neural network behaviors.
+ the connection with generalization of neural networks is an interesting observation.

Weakness:

- lack of important details on the experiments and related work. Despite briefly introduced the baselines in the Appendix, the main text should be self-contained. Why are these baselines chosen is also not clear. Without a clear description of the baselines, experiments, or codes greatly hinders the reading and evaluation of the results.
- limiting novelty: the concept of computing code length is not news in deep learning (Blier et al., 2018; Kim et al., 2018), and the authors are only incrementally extending the complete formulation of information transfer in Zhang et al., 2020 into the expected codelength. It is not clear the contribution, both theoretically and empirically.
- why is eq 10 a valid assumption? if not, the entire basis for this measure is not solid.
- missing definition of math notations: e.g. what is f_square in eq 14? the technical sections are below standard to top-tier conferences.
- didn't show a clear benefit over existing methods: section 3.2 explored the invariances of the measure in different parameterization settings of neural networks. however, we see that many baselines are equally invariant if not better than the proposed measure.  even in section 3.3, we see EMD performing just as well in capturing the change in training process.
- lack of comparison with other methods in Fig 4. It is likely that other similarlity measures offers interesting visualizations like this in three-dimensional space as well. Why is this method better in any way?
- several acronyms used without introducing their definition first.



Suggestion:

This submission appears to only preliminary work to a potentially interesting approach to understand the behaviors of deep learning.

The writing is a little bit hard to follow, mostly due to a lack of details in the empirical evaluations from section 3 onward. I think it would require some significant rewriting before ready for publication.

The contribution of the work is also a bit confusing. It feels like the heavy lifting of the idea actually comes from the formulation of information transfer in Zhang et al., 2020. The empirical study also didn't show any clear benefit of the proposed approach over existing ones.

---

> ### Author Response · Authors · 2020-11-20
> **Response to Reviewer3, more details about experiments and baselines, quantitative measurements of the advantage of the proposed method, improvements in math and writing**
>
> Thanks for your comments! These are very helpful advices that we updated the paper to addressed each:
>
> Details about baselines: we have expanded Appendix B to provide formulas to calculate each baseline measure that is used in the paper. We also added short descriptions of these baselines when introducing them in Figure 1, along with the reason to choose them as baselines. The reason for choosing the baselines are mainly 1) to include distance measures in different spaces (parameter space, representation space, and output space), and 2) to include common and straightforward measures like L2 and cosine, and more sophisticated measures like EMD and CKA that are shown to have invariance properties.
>
> Details about experiments: Section 3.1 is modified to avoid possible confusion about experiment setting, and we added the model architecture and datasets used for training $\theta_A$ and $\theta_B$ in Section 3.3.
>
> Related work: thanks for pointing out the missing reference. We added them in the introduction.
>
> Novelty: the main novelty of this paper is 1) propose a calculable information distance using prequential codelength as approximations of Kolmogorov complexity, 2) use KL-divergence in prequential training and coding to directly estimate the expectation of codelength, which avoids the sampling process and improved sample efficiency, and 3) presents an empirical study that shows the invariance, utility, and universality of the approximated information distance measure.
> In the original submission, 2) was only in the Appendix. After we revised the theoretical derivations in Section 2 with comments from one of the reviews, we arrived at new results that more closely match the empirical observations. This allow us to reclaim 2) as a contribution of this paper.
> For 3), we explain in the next bullet point:
>
> Clear benefit of the proposed method: in this revised version of the paper, we added quantitative results for invariance experiments in Section 3.2, which more clearly shows the advantage of the proposed measure $d_p$. In section 3.3, we also added two tables to compare whether each method shows the correct trend and is able to distinguish different models. These show that $d_p$ has better invariance than all other baseline methods, and is the only one measure that has the correct trend and identifies different distance models in both scenarios.
> For results in Section 4, these application experiments are mainly used to show the utility and universality of the proposed measure $d_p$. While we believe methods like CKA can also give meaningful results in experiments of Section 4.1, no baseline method can directly apply to all four experiments in Section 4. This is because information distance is universal and does not depend on the choice of model, or that a model must have meaningful representations as CKA implicitly requires.
>
> Math: we have re-written the derivations in Section 2.2 to make it more solid. Equation 4 is more of a theoretical concern that shows we should train $f_A$ to converge to $f_B$ for exact results. In practice because we have limited examples in the dataset, we seldom can train $D_{KL}( f_A||f_B)$ to very small. Therefore, we focus on using empirical results to show that while the approximations lack a solid convergence guarantee, it can still serve as a useful measure.
> Missing definition of symbol $\square$ in (11) is also added.
>
> Acronyms: we have added full names of acronyms like EMD and CKA in the description of Figure 1, where they first appear.

---

### Official Review · AnonReviewer4 · 2020-10-28

**Rating:** 4
**Confidence:** 3

**Review:**

1, Summary of contribution:
The paper proposes a new distance measure between a pair of neural networks, which is invariant to the reparameterization. Specifically, it introduces prequential coding to approximate the Kolmogorov complexity between two neural networks, and
The paper also conducts empirical studies on the properties of the proposed distance. (especially the relationships to model diversity and generalization in the section 4.3)


2, Strengths and Weaknesses:
The idea of introducing Kolmogorov complexity is interesting, and many interesting ablation studies are done.  At the same time,  there seems to be a mistake in the important step of their theoretical justification.  Also, although the paper presents many empirical studies,  it does not extensively compare the proposed distance to other distances on a theoretical basis, and this makes it difficult for the readers to understand the position of this study in the field.

3, Recommendation:
Because the validity of  “distance” in an application is rather subjective and depends on the tasks,  the significance of a study in this field most likely hinges on theoretical soundness and the depth of theoretical analysis. Because there seems to be a flaw in the theoretical soundness, I suggest rejection.

4, Reasons for Recommendation:
There appears to be a mistake in the equation (4)-(6), which is the core of the justification of the proposed distance.
In equation (4)-(6),  the authors endeavor to take an expectation with respect to {(x_i,  y_i)}.  However, in the transition from LHS of (4) to  RHS of (4), the authors seem to be forgetting the fact that theta_i is also a variable of {(x_m, y_m) ; k =1,...i-1}.
In other words,   E_{x_{0:i}, y_{0:i}} [ log p_{theta_i} (y_i | x_{1:i},  y_{1:i-1}) ]  is not   E_{x_{i}, y_{i} } [ log p_{theta_i} (y_i | x_{1:i},  y_{1:i-1}) ]
On this basis, I do not believe that (4) can be simplified into (6).

If the problem raised in 4 can be resolved, I would very much like to change my rate. Please let me know if there is some misunderstanding on my part.

Also, because the theoretical analysis is an important part of this research, more theoretical studies on the properties of the proposed distance.

\
---Post rebuttal---

Thank you very much for the response, and I understood that some of the concerns I raised can be resolved empirically under appropriate conditions.   After the response, however, I think I am still comfortable with the original score, because this is (as I understood it) a theoretical paper and I felt that it is necessary to make a judge based on theoretical solidness.

---

> ### Author Response · Authors · 2020-11-20
> **Response to Reviewer4, corrected derivations and included new insights in the paper**
>
> Thanks very much for pointing out the expectation over $y_{1:i-1}$ that we missed in equation 5-6, which is indeed central to the derivation. We have corrected this error and revised the equations, and now the new derivation leads to a closer connection to the empirical method we used in experiments. Actually, this helps us gain insight into why using KL-divergence in both prequential training and coding works. This method of estimating expected codelength without sampling is what we think another contribution of this paper. This correction eventually makes the work more complete. We have re-written Section 2 and Appendix A to reflect this new observation.

---

> > ### Comment · AnonReviewer4 · 2020-11-21
> > **Response**
> >
> > Thank you very much for the reply.
> > However, I still do not understand how the argument from (13)  ~ (17) in the appendix would suggest that the second term in (13) is small. In particular, if i is small, intuition dictates that  (13) cannot be small. Is it possible to provide a more concrete argument with bounds?

---

> > > ### Author Response · Authors · 2020-11-25
> > > **New empirical results for Equation (13)**
> > >
> > > Thanks for pointing out the potential problem for small i. The second term in Equation (13) (which we denote by $\Delta$) might be difficult to bound without any assumptions on the distribution of y and the model $\theta$, hence we resort to empirical evidence and present empirical values of $\Delta$ in Table 2 of Appendix A. In our experiments, $\Delta$ is often very small compared to the first KL-divergence term, regardless of i. For very small i (e.g., i=2), $\Delta$ could be larger. However, very small i is not encountered in practice, because prequential coding for neural models is usually done in a “block-wise” fashion (see Section 3.4 of [1]) to save computation time. As a result, leaving out $\Delta$ in (13) is insignificant for the codelength we are calculating.
> > >
> > > Also about theoretical comparisons to other distances: on a theoretical basis, our proposed distance is based on Kolmogorov complexity in the hypothesis space H, which is quite different from other distances that study Shannon information based quantities in the input and output space, and is even more different from distances in the parameter or representation space. One reason that comparisons are hard to make theoretically, is the lack of analytical results for Kolmogorov complexity on complex models like neural networks. In this regard, we believe our method's position is quite unique in the relevant studies of distances and geometries of neural models. In this paper, we mainly focus on providing a theoretical formulation and a practical method, and use empirical evidence to show its effectiveness. More rigorous theoretical results are necessary for more in-depth understanding of the method, which we think are important future work.
> > >
> > > [1] Blier, Léonard, and Yann Ollivier. "The description length of deep learning models."

---

### Official Review · AnonReviewer2 · 2020-10-30
**Prequential coding based distance measure for neural networks**

**Rating:** 6
**Confidence:** 4

**Review:**

This paper explores the problem of designing a distance measure in the space of the functions parameterized by neural networks. Ideally, such a measure should be independent of the parameterization of the networks. Also, the measure should support quantifying the distance between the networks with different structures and/or different underlying training tasks. The information distance meets this natural requirement. However, information distance is computational infeasible as it is defined by Kolmogorov complexity.

This paper utilizes the length of prequential coding to approximate the information distance which leads to a practical distance measure for neural networks. The paper empirically verifies that this distance measure is indeed invariant to the parameterization of the network. Furthermore, the paper shows that the distance measure explains/captures various behaviors of neural networks, e.g., the effect of regularization, connection between model diversity and performance for network ensembles, and generalization.

The paper studies an interesting and timely problem. The empirical results demonstrate the value of prequential coding based distance measure for understanding the behavior of neural networks. That said, there are some concerns about the novelty and presentation of the paper:
1. Prequential coding has been previously used in the context of neural networks, e.g., [1], which limits the novelty of this paper.
2. There is significant room for improvement in term of the presentation of the paper:
   i) Consider organizing the text in Section 2 (before Section 2.1) in various subsections. This would help streamline the presentation. E.g., prequential coding can be introduced in a subsection. Similarly, it is not clear why the discussion on information transfer measures is necessary after (6). Perhaps, the authors can introduce (11) after (6) and then talk about information transfer measures in a subsection.
    ii) The last two paragraphs of Section 2.2 are not very clear. Please consider expanding those paragraphs to make the message more clear.
    iii) Please use the expansion of EMD and CKA the first time these terms are used.

######## Post rebuttal ###########

Thank you for your response and the efforts to improve the presentation of Section 2. After going through other reviewers' comments and the authors' responses to those, I am comfortable with my original score.

---

> ### Author Response · Authors · 2020-11-20
> **Response to Reviewer2, clarified novelty of the proposed method and improved presentation of the paper**
>
> Thanks for your comments which help us to improve the manuscript. We have revised the paper regarding concerns about novelty and presentation:
>
> Novelty: the main novelty of our method lies in 1) using prequential coding to approximate conditional Kolmogorov complexity in information distance, and 2) using KL-divergence in prequential training and coding to directly estimate the expectation of codelength, which avoids the sampling process and improved sample efficiency.
>
> In the original submission of the paper, we put 2) mainly in Appendix A. Although we used 2) in experiments and it worked well, at the time of submission we did not have a concrete derivation for 2) mainly because of an error in equation (5) (as pointed out in one of the reviews). In the current updated version of the paper, we have revised the derivation in Section 2 and added discussion on 2), in which we claim 2) as a contribution of this paper.
>
> Presentation: we have taken the advices and significantly rewritten Section 2, which now have a more logical division of topics into subsections. Less central comments like (8) are moved to later in the text. Expansions of terms such as CKA are added. For discussion about data-dependency, we have already run out of the extra page to expand in the main text, so we added further explanations about the final claims in Section 2.4 in Appendix A.

---

### Decision · Program_Chairs · 2021-01-07
**Final Decision**

**Decision:**

Reject

**Comment:**

All the reviewers agree that the paper studies an important and interesting problem. However the reviewers felt the paper is still in preliminary stages, with incorrect derivations, missing comparisons/references,  and writing. While the authors updated the paper during the discussion stage addressing some of the concerns, the paper still needs more work in adding appropriate comparisons and in presenting the concepts more clearly. Hence I believe the paper is not yet ready for publication and encourage authors to continue their work.